# Assessing the sustainability of yellow anaconda (*Eunectes notaeus*) harvest

**Bruno F. Camera**[1]*, **Itxaso Quintana**[2], **Christine Strüssmann**[1], **Tomás Waller**[3‡], **Mariano Barros**[3‡], **Juan Draque**[3‡], **Patrício A. Micucci**[3‡], **Everton B. P. Miranda**[4]

**1** Programa de Pós-Graduação em Zoologia, Instituto de Biociências, Universidade Federal de Mato Grosso, Cuiabá, Mato Grosso, Brazil, **2** Biodiversity and Development Institute, Cape Town, South Africa, **3** Fundación Biodiversidad Argentina, Buenos Aires, Argentina, **4** Programa de Pós-Graduação em Biodiversidade e Agroecossistemas Amazônicos da Universidade do Estado de Mato Grosso (UNEMAT), Alta Floresta, Brazil

☯ These authors contributed equally to this work.
‡ TW, MB, JD and PAM also contributed equally to this work.
* camera_bruno@hotmail.com

## Abstract

Sustainable wildlife management is necessary to guarantee the viability of source populations; but it is rarely practiced in the tropics. The yellow anaconda (*Eunectes notaeus*) has long been harvested for its leather. Since 2002 its harvest has operated under a management program in northeastern Argentina, which relies on adaptive management practices, that limit the minimum body length permitted for harvesting, the number of active hunters and the length of hunting seasons. Here we investigated the effects of yellow anaconda harvest on its demography based on 2002–2019 data and show that exploitation levels are sustainable. The gradual reduction in annual hunting effort, due to a decrease in the number of hunters and hunting season duration, reduced the total number of anacondas harvested. Conversely, captures per unit effort increased across the study period. The body size of anacondas was not influenced by the harvesting, and more females than males were caught. We also found that a decrease in mean temperature positively influenced anaconda harvest and the capture of giant individuals. Because sustainable use is a powerful tool for conservation, and anacondas are widespread in South America, these discoveries are highly applicable to other species and regions.

## Introduction

Ever since the Paleolithic [1], humans have strived to make raw animal skin into leather, a resource that has been incredibly useful throughout human history, having grown into a US $90 billion/year industry in the world today [2]. Due to its remarkable economic value, leather became an important driver of wildlife management programs [3], as seen in the sustainable management of alligators (*Alligator mississipiensis*; [4]) or the poaching of Asian elephants (*Elephas maximus*; [5]). This management has generated a still growing body of knowledge

**Data Availability Statement:** The technical reports with the original data used in this study are issued by Fundación Biodiversidad and can be requested by any interested person. However, they do not

represent third part data. Additionally, we provided a copy in Mendeley repository: https://data.mendeley.com/datasets/x8wwf3hzxb.

**Funding:** Conselho Nacional de Desenvolvimento Científico e Tecnológico (CNPq) offered financial support through public call number 003/2014 (process 155536/2014 and 456497/2014–5) and through the research fellowship 3123038/2018-1. Fundação de Amparo à Pesquisa do Estado de Mato Grosso (FAPEMAT) conceded a grant through the public call 017/2015. The funders are Brazilian and had no role in study design, data collection and analysis, decision to publish, or over the management plan, which is an Argentinean endeavor.

**Competing interests:** The authors have declared that no competing interests exist.

[6, 7] that is vital for the integration of sustainable wildlife management into local economies, which is a key trait of proper biodiversity stewardship [8].

Reptile skin has long been considered a major fashion product due its strength and diversity of patterns [9]. Reptile flesh, eggs [8] and other subproducts [10] also have nutritional and medicinal value for traditional human communities [11]. Commercial hunting of Neotropical reptiles dates from the beginning of the 18th century, starting as a source of oil [12], with the fashion industry becoming a notable driver of wild reptile harvest in the 1920s. This practice initially occurred in Southeast Asia, quickly expanding to Africa and South America in the following decade [13]. Currently, the snake families Pythonidae and Boidae dominate the global trade in reptile skin [14] due to their large sizes. In South America, snakes of the genera *Eunectes*, popularly known as anacondas, are exploited for this international trade [13].

Despite being a historically significant practice, harvesting reptiles is prohibited in several countries and sustainable management practices are seldom proposed as pathways to legalization [15, 16]. However—when outlawed—the continuous illegal use of wild meat, leather, or eggs contributes to the marginalization of traditional communities that rely on these resources [17]. At the same time, reptilian wildlife suffers the consequences of irresponsible use, such as the killing of undersized or immature individuals, which can lead to population crashes [18]. This is specially problematic in remote localities, where such practices are often economically irreplaceable [19]. While the uncontrolled commercial harvesting of reptiles is considered one of the main threats to their conservation [20], the creation of management plans can help to ensure the economic viability of conservation initiatives, reducing their costs [21–23] and promoting the recovery of biodiversity [16] and of managed populations themselves. Moreover, management plans focused on the sustainable use of wild resources can help ensure social and environmental quality [24, 25]. Thus, the implementation of responsible management plans should be a priority in any viable conservation agenda.

The creation and execution of wildlife management plans are challenging, as there is often an absence of technical or scientific criteria available for decision making, well-trained human resources, studies of supply chains, marketing strategies, specific parameters for each taxon [24], and inadequate remuneration for hunter communities [26]. In some cases, as for the Central American river turtle (*Dermatemys mawii*), the laws and regulations for the management of the species are simply ignored by locals [27]. Nevertheless, examples of Neotropical reptiles being successfully managed are available, as is the case for the river turtles *Podocnemis expansa* and *P. unifilis* [16, 28], caimans (*Melanosuchus niger*, *Caiman crocodilus*) [24], *C. yacare* [29], and *C. latirostris* [30, 31], lizards (*Salvator merianae* and *S. rufescens*) [32] and snakes (*Boa constrictor* [33] and *Eunectes notaeus* [34, 35]).

Yellow anacondas are managed in Argentina by *Fundación Biodiversidade Argentina* since 2002. Because management plans must ensure the continuous availability of the exploited resource [36], a set of biological and managerial parameters were selected to guide this management project [35]. During a pilot period (2002–2004), the managers established that accredited hunters should be restricted to: 1) hunting during an annual hunting season, and 2) only capturing individuals longer than 200 cm in snout-vent length (SVL), which correlates to a 230 cm dried skin. However, a small harvest of undersized animals has also been tolerated. The proportion of undersized skins harvested can help to elucidate the levels of misconduct among accredited hunters, and to track if this willingness to bend established rules is being affected by factors such as the increase in required hunting effort.

The 230 cm size threshold guarantees that most individuals are able to breed, and allows anacondas to be managed from an effort perspective—instead of using quotas from a known population—since estimating snake populations through capture, mark and recapture is unfeasible [35]. After skins have been extracted and dried by hunters, they are sold to a local

buyer, where the *Programa Curiyú* team collects and systematizes harvest data. Since it is not possible to confidently predict the response of the yellow anaconda population to this harvest, the program uses an adaptive management framework [37], which allows for periodic changes in management parameters such as hunting effort, hunting season duration, and minimum size of individuals. Such variables have been occasionally redefined as field data was produced and analyzed [35, 38].

Given that the essence of biological monitoring lies in data comparison over time [39], in this work we conducted a comprehensive historical analysis of the *Programa Curiyú* database in order to investigate the sustainability of anaconda harvesting. We evaluated historical trends of: 1) capture rates; 2) total number of anacondas harvested each year; 3) mean SVL of skins; 4) mean SVL of giant skins, which correspond to the 5% largest individuals captured each year; and 5) proportion of skins with SVL > 230 cm. These demographic parameters were used due to the cryptic nature of snake populations [40, 41], which cannot be measure through other traditional methods like capture, mark and recapture [14].

Each monitored parameter reveals something different about the demographic cycles of a snake population. Capture rates can inform how hunting efforts vary annually, and they are expected to increase if the population is dwindling [42]. Body size is positively correlated with fertility in female snakes [15, 43], and information on their average and extreme sizes reveals the effects of both hunting and other factors (e.g. catastrophic droughts, predation, or reduced prey abundance can reduce snake sizes; [44, 45]) over the population. Proportion of undersized skins by its time allow learning the proportion of misdoing and how this is affected by other factors, such as increase in the required hunting effort [7, 39, 46]. Together, these parameters allow us to better understand anaconda populations, just as they do for other species under older and more well-established management initiatives, such as the American alligator [47].

Finally, because harvest is dependent on climatic conditions due to the thermoregulatory behavior of anacondas, we analyzed the influence of mean air temperature and water levels on hunting productivity. By testing sustainability using the forementioned parameters, we can identify management weaknesses and strengths, help propose new parameters for management and identify marginal or suboptimal harvest levels. All these results can help improve conservation policies for the Earth's largest snakes.

## Methods

### Management program

Ministerio de la Producción y Ambiente de la Provincia de Formosa (Formosas' Environmental Bureau) provided the licenses for the management program as a whole. No animals were killed for the specific purposes of our study. Local communities in northeastern Argentina have historically hunted yellow anacondas—locally known as *boa curiyú*—for their skin [34]. Between 1980 and 1999, around 320,000 individuals were harvested [48], but the harvest was terminated abruptly when trade was effectively banned in 1999, affecting the subsistence of local residents [34]. In 2002, an adaptive management program for the species was developed [48] and remains active as of the publication of this research, harvesting ~6500 individuals in 2022.

The first problem in the implementation of an anaconda management program is the gathering of biological data, due to the cryptic nature of snakes [15]. Traditionally, data on population size and structure are first obtained, either by direct counting or by the use of abundance indices, so that hunting quotas can be defined [24, 28, 49]. However, this traditional approach

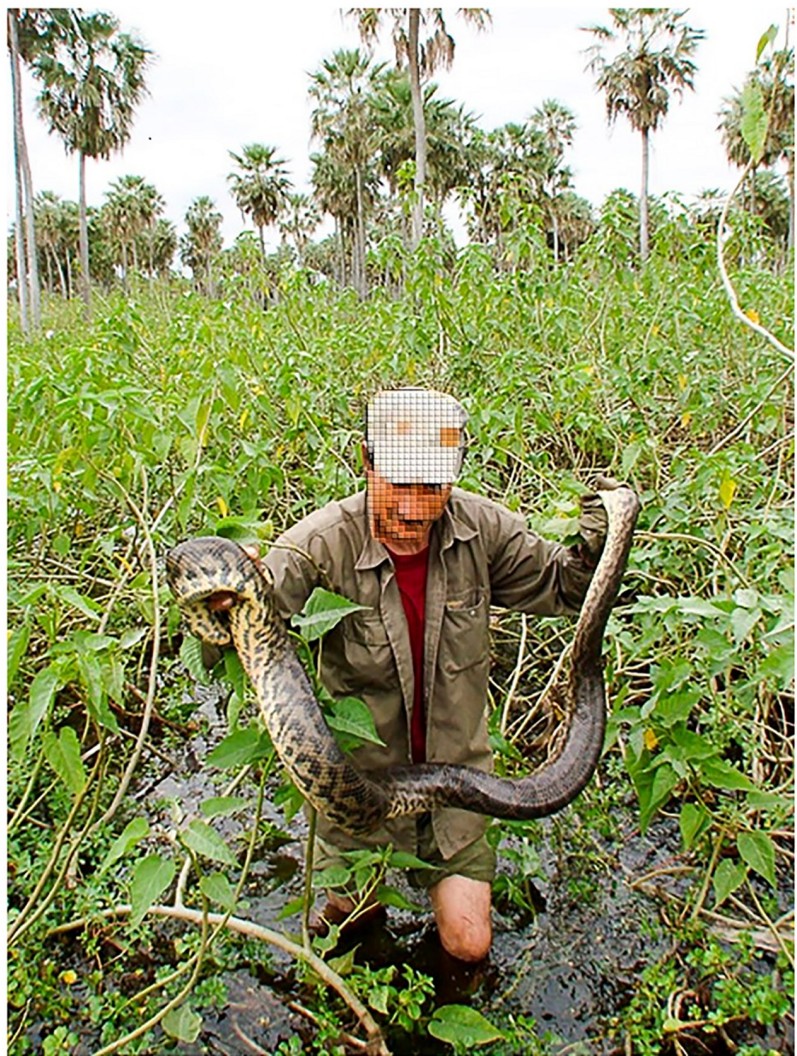

**Fig 1. A hunter capturing an yellow anaconda (*Eunectes notaeus*) by hand in La Estrella marsh, northeastern Argentina.** Hunting is selective and undersized individuals can be released unharmed. Photo by Programa Curiyú.

was considered unrealistic for yellow anacondas, and the *Programa Curiyú* performed a pilot study to define which parameters were important to guide the program under the adopted adaptive management approach [34, 37].

The main goal of adaptive management is to provide a decision-making scheme that can be remodeled as new information becomes available [50], like an experiment capable of learning from itself [51]. From a practical perspective, *Programa Curiyú* controls the harvest by limiting hunting effort (number of hunters and hunting season duration) and restricting hunts to manual capture (Fig 1). As a result, there is no direct control over the number of anacondas harvested, and the managed population size is inferred from capture rates [34]. Additionally, previous studies, via estimates of yellow anaconda reproductive biology, have established a minimum SVL of 230 cm for captured individuals (representing a snake of ~200 cm; Waller et al. 2007).

## Study area

*Programa Curiyú* takes place in La Estrella marsh (24˚08'S, 60˚35'W), a seasonally flooded section of the La Plata River basin, and in the Chaco vegetative domain of Formosa province, northern Argentina. The marsh arose as a result of the natural sedimentation of the Pilcomayo River, which led to the natural damming of its course in 1960, creating a marshland of about 3,000 km$^2$ [52]. The landscape is characterized by the macrophytes that cover the water and the epiphytic vegetation that covers dead hardwood trunks—locally known as *champas*—with an emergent strata formed by caranda palms (*Copernicia alba*, Arecaceae) [52].

In this subtropical region, the climate is characterized by extreme seasonal fluctuations between flood (January-May) and dry (October-December) periods. The marsh's flooding pattern, from December to February, is related to the Andean rainfall. The water level progressively drains out around April/May, with minimum water levels or total drought occurring from October to January. During winter, the area is subjected to cold fronts, when temperature drops for few days. Between 1961–1990, minimum temperature (12.1˚C) was recorded in July, during winter, and the maximum average monthly temperature (33.4˚C) was recorded in January during summer [53].

## Study species

Yellow anacondas (*Eunectes notaeus* Cope, 1863) are non-venomous semiaquatic Boidae snakes that remain active throughout the year [35]. During summer at La Estrella (December-March), they hide in moist aquatic vegetation, which makes them difficult to detect. During the winter, however, and especially after cold fronts, they can easily be found basking in the sun on the *champas* [35] (Fig 2). Yellow anacondas exhibit fast growth and remarkable sexual dimorphism—females are 20% longer and twice as heavy as males [35]. Reproductive activity begins in the spring, with parturition (1♂: 1♀) occurring in early autumn, in April [35]. Females reach maturity when they are around 2 m of SVL, produce litters of 7–42 young every other year, and are usually maturing their eggs internally during the hunting season [35]. Yellow anacondas are generalist predators that both ambush and actively hunt for prey [54, 55]. They feed on invertebrates, fishes, reptiles, mammals, birds, eggs, and eventually carrion [35, 54–57].

## Data collection

We obtained data on yellow anaconda hunting activities from the technical reports issued annually by the *Fundación Biodiversidad*. This institution created and currently supervises the *Programa Curiyú*. We emphasize that no anacondas were killed solely for the purpose of the current study. We used data available for the period of 2002–2019. However, in 2013 a severe drought affected the flooding regime at La Estrella [38, 58]. Consequently, there was no hunting season that year, and 2013 was excluded from all analyzes.

A wide variety of parameters were calculated for the 17 years of harvest. Because overharvesting can reduce resource availability [39, 59], we analyzed the total number of anacondas harvested each year, as well as captures in the most productive month (July, the month with the highest number of captures in each hunting season). Overharvesting can also affect the sex ratio and average body size of harvested populations [60, 61], so we also analyzed the proportion of captured females, mean SVL of skins, mean SVL of skins > 230 cm, and the proportion of skins with SVLs > 230 cm. Additionally, we calculated the mean length of giant skins (i.e., top 5% of the longest skins from each year). Because skins are usually stretched by hunters to

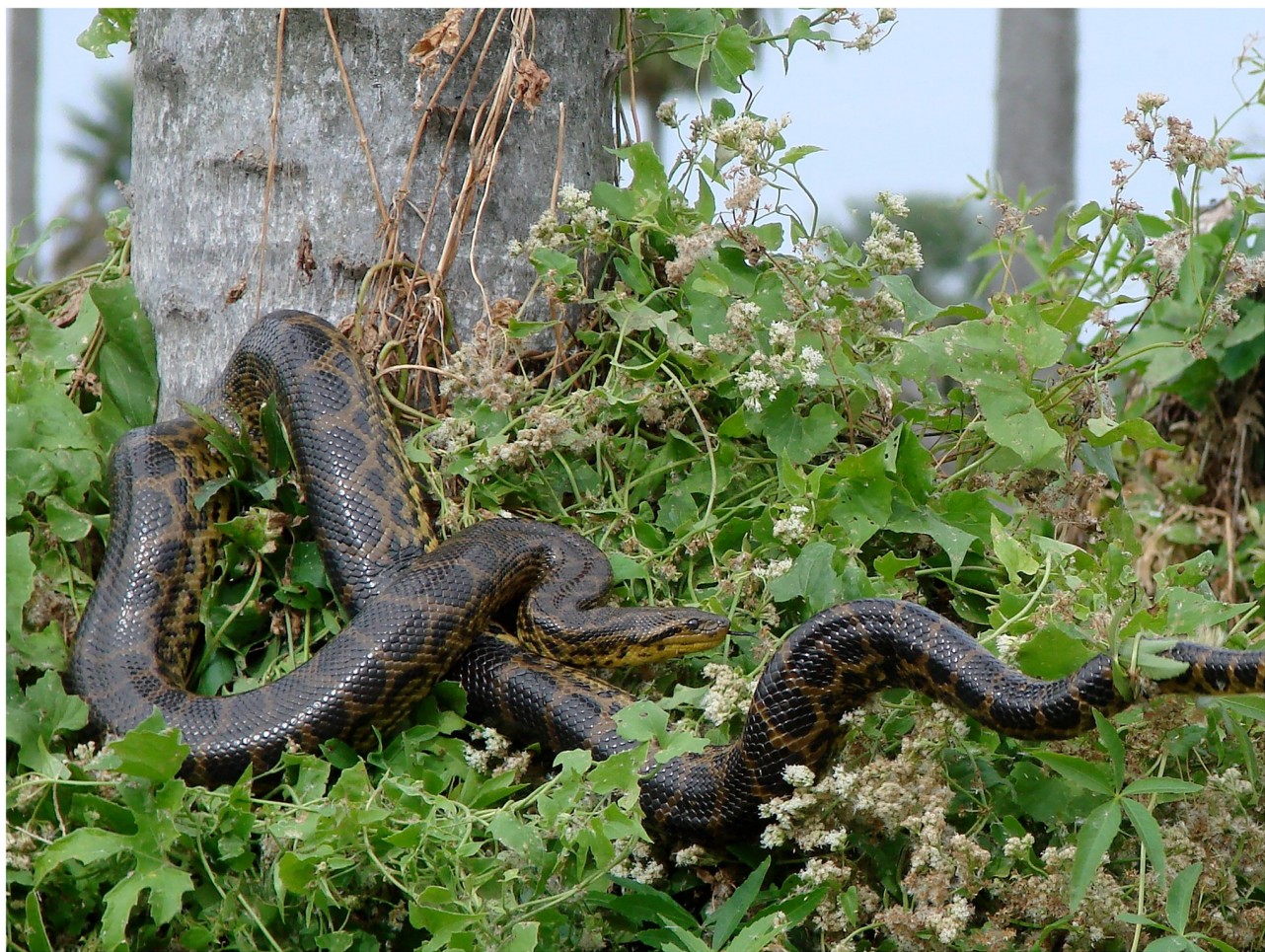

**Fig 2. Yellow anaconda (*Eunectes notaeus*) basking in the sun on the *champa* at La Estrella marsh, northeastern Argentina.** Photo by Programa Curiyú.

dry them out, we obtained the true SVL values using the following Eq (1) [49]:

$$\text{SkinSVL} = \frac{\text{raw skin SLV} + \text{raw skin width} \times 10}{2} \qquad (1)$$

Furthermore, we estimated the SVL of living snakes according to the following Eq (2) [48]:

$$\text{LiveSVL} = 11.71 + 0.66 \times \text{raw skin length} + 1.59 \times \text{raw skin width} \qquad (2)$$

We also analyzed the number of accredited hunters active during the hunting seasons, total days of harvest, and capture rates (captures per unit effort; CPUE), determined by the Eq (3):

$$\text{CPUE} = \frac{c}{n \times d} \qquad (3)$$

Where: $c$ = total number of anacondas harvested in year x; $n$ = number of accredited hunters in year x; $d$ = hunting season duration (days) in year x [49]. We also calculated the CPUE of the most productive month (July). The nominal effort is not homogeneously distributed across the whole hunting season: not all hunters begin their activities when the start of the season is

announced and falling water levels can make hunting especially difficult within the marsh towards the end of the season. To circumvent this issue, we analyzed the effect of the difference between capture rates in July and in the remaining hunting season.

Data on environmental variables was also collected, such as mean hunting season temperature (ºC) and maximum water level in the marsh. Air temperature data was collected by the *Programa Curiyú* every two hours, from 2003 to 2018, using data loggers in a shaded station located in Fortín Soledad—a small city on the southeastern edge of La Estrella marsh. Air temperature data for 2019 was obtained from World Weather Online [62] for Las Lomitas (located 64 km away from Fortín Soledad). Water level records were taken by the local police department between 2002–2016 using a measuring gauge located on the edge of Fortín Soledad. This means that the water level was recorded once the water arrived at the edge of the city and reached approximately 0.4 m in the measuring gauge. However, the *Programa Curiyú* estimates that this is equivalent to 1.1 m in the deepest part of the marsh. Because there are no continuous records of the water level throughout the year due to seasonal ebbing, we used peak water level as a proxy for water volume. During the two driest years, 2005 and 2013, flooding was so minimal that the water never reached the measuring gauges at Fortín Soledad, and there were no water level records for these years.

## Data analysis

We used four explanatory variables: 1) year (to analyze the trends of the different hunting parameters); 2) hunting effort; 3) mean temperature; and 4) water level in the marsh during the hunting season. Hunting effort was calculated by multiplying the number of hunters and days of hunting. Mean temperature during the hunting season was calculated using daily temperatures between May and September, which correspond to the austral winter.

To analyze historical trends in anaconda harvesting parameters we carried out Generalized Additive Models (GAMs), using the 'mgcv' package [63]. This approach allows the shape of the relationship between the response and explanatory variables to be determined by the data, rather than follow a prescribed functional form [64]. We built different models for each response variable, which are: total captures, number of hunters, hunting days, total CPUE, CPUE in July, proportion of captures in July, proportion of captured females, skin SVLs, skin SVLs > 230 cm and proportion of skins > 230 cm. The variables skin SVLs and skin SVLs >230 cm were log transformed. We considered the whole duration of the project (from 2002 to 2019) in the analysis of these trends, including the first two years when the pilot study was carried out.

We used GLMs to assess the effects of the following predictive variables over the harvested population: mean temperature, hunting effort, and year. The explanatory variables were the harvested population parameters: total captures, mean SVL, skin SVL > 230 cm, and SVL of giant skins. Anaconda hunting was extended beyond the winter season during the pilot study, therefore, we exclude the first two years (2002–2003) from these specific analyses.

We standardized and centralized the explanatory variables to facilitate the interpretation of the estimated coefficients. Because year and hunting effort are highly correlated (r = -0.72), we built separate models to incorporate both variables [65]. Among these, the best model was selected based on the lowest value of Akaike Selection Criteria (AIC; [53]). AIC estimates prediction error [66], and can consequently test the quality of different statistical models for different datasets [67]. We performed all analyses in R software v.3.5.1 [68] and we used a significance level of 5%.

Poisson distributions were used to model captures, number of hunters and hunting days at first, but when overdispersion [69] was observed we used a negative binomial distribution [70]

as a corrective [71]. To analyze mean SVL of skins and CPUE, we used a Gaussian distribution; a Shapiro-Wilk test was used to assess normality [72]. We modelled the proportion of captures in the most productive month, the proportion of skins with SVL > 230 cm, and the proportion of harvested females using a binomial distribution, and used the function 'glm.binomial.disp' from DISPMOD package [73] to account for overdispersion. We fitted quadratic relationships when such need was suggested by exploratory graphs. Finally, to understand the effect of the different temperatures in July and in the rest of the hunting season on capture rates, we modelled the relationship between CPUEs and mean temperatures using a Gaussian distribution.

## Results

### Trends in harvest parameters

A total of 64,176 yellow anacondas were harvested during the 18-year existence of the *Programa Curiyú* (Fig 3).

We observed a decrease in the number of hunters, as well as in the hunting season duration (Fig 4a and 4b; see S1 and S2 Figs in S1 File), after the first two years. The current average number of hunters is 182 (± 128), while in 2002 and 2003 they numbered 473 and 450,

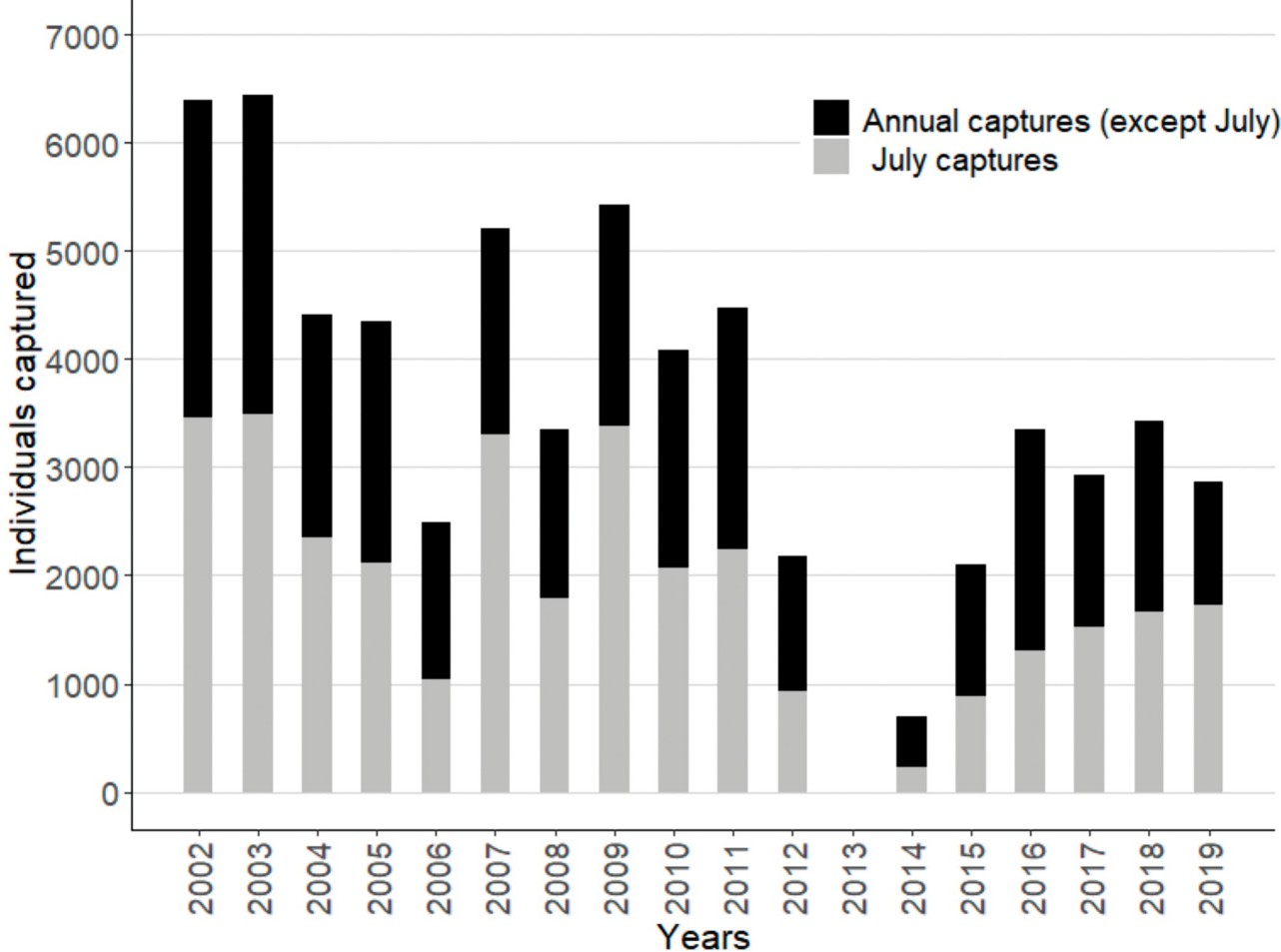

**Fig 3. Yellow anacondas (*Eunectes notaeus*) harvested by the *Programa Curiyú* between 2002–2019 in northeastern Argentina.** Total annual captures and captures in the most productive month (in grey) are shown.

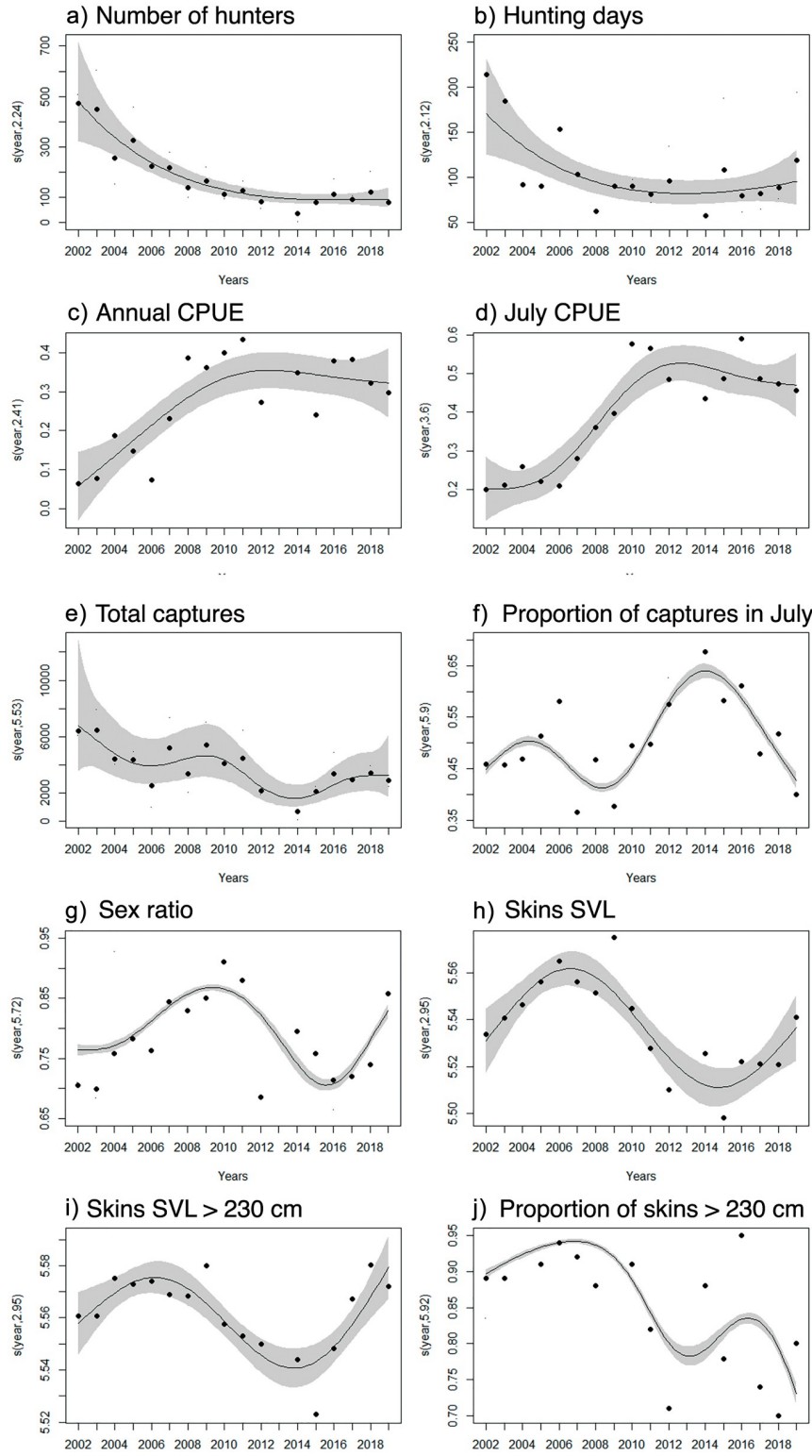

**Fig 4. Smoothed trends of yellow anaconda (*Eunectes notaeus*) harvest parameters measured by the *Programa Curiyú* between 2002–2019 in northeastern Argentina.** The y-axis represents the smooth term and the EDF value, and the grey area indicates the 95% confidence interval obtained from the Generalized Additive Models (GAMs).

respectively. The hunting season stretched for 105 (± 42) days during 2004–2019, while in 2002 and 2003 they lasted 214 and 185 days, respectively. On the other hand, we observed an increase in capture per effort unit—more anacondas were being captured per unit of time (see S3 and S4 Figs in S1 File). Annual CPUE increased until around 2008 and then stabilized (Fig 4c), while the CPUE in July increased until 2010, stabilizing from then on (Fig 4d).

Because of the decline in hunters and duration of the hunting season, the total number of captured anacondas also fell between 2002 and 2019 (Fig 4e; see S5 Fig in S1 File). The highest numbers of anacondas were hunted during the pilot phase of the *Programa Curiý*, in 2002 and 2003 (6391 and 6442 individuals, respectively), and the lowest in 2014 (697 individuals; Fig 3). Annual captures and captures in the most productive month (July) were highly correlated (r > 0.9). On average, 50.1% (± 8.4%) of the anacondas were captured in July, but this proportion varied across the years (Fig 4f). In general, more females than males were captured every year (1♂: 4.6 (± 2.1♀), with the highest proportion of captured females taking place in 2010 (1♂: 10.1♀; Fig 4g).

The size of captured individuals oscillated over time, and it was highest in 2009, with a mean SVL of 263.76 cm, decreasing in the following years until the lowest mean SVL was recorded in 2015 (244.21 cm; Table 1), with the average size of captured individuals increased slightly between 2016 and 2019 (Fig 4h). A similar pattern was observed for the mean SVL of skins longer than 230 cm (Fig 4i). Mean SVL > 230 cm was high until 2009, but then decreased over time until 2015, when it reached its lowest value. This mean increased vertiginously in the following years, reaching its highest mean value in 2019 (Table 1; Fig 4i). The proportion of skins with SVL > 230 cm was high across the study period 86.3%, (± 8.8%), but it decreased as years passed (Fig 4j; see S6 Fig in S1 File). Moreover, the mean SVL of giant individuals (top 5%) slightly decreased between 2004 and 2016 (see S7 Fig in S1 File), increasing again in 2016, when it returned to original values (Table 1). The estimated size of the longest yellow anaconda was 410.71 cm of SVL, from an individual captured in 2009 (S8 Fig in S1 File). Selected GAMs for each harvest parameter are summarized in Table 2.

**Table 1.  Biological attributes of yellow anaconda skins harvested under the *Programa Curiyú*.** Mean skin width, mean snout-vent length (cm), and mean snout-vent length of skins > 230 cm are shown, as well as the number and % of skins with snout-vent length > 230 cm of anacondas harvested between 2002–2019. The mean snout-vent length (cm) of giant skins (corresponding to the 5% longest skins) is shown for the 2004–2016 period.

| Year | Mean skin width (cm) | Mean skin SVL (cm) | Mean skin SVL >230 cm | Number (%) of skins with SVL>230 | Mean 5% longest skin SVL (cm) |
|------|------|------|------|------|------|
| 2002 | 24.64 | 253.07 | 260.00 | 5688 (89) | NA |
| 2003 | 20.01 | 254.83 | 260.00 | 5733 (89) | NA |
| 2004 | 24.06 | 256.32 | 263.75 | 4325 (98) | 345.42 |
| 2005 | 26.18 | 258.82 | 263.22 | 3961 (91) | 341.99 |
| 2006 | 26.23 | 261.1 | 263.46 | 2344 (94) | 339.47 |
| 2007 | 25.91 | 258.8 | 262.16 | 4794 (92) | 344.07 |
| 2008 | 25.96 | 257.59 | 262.01 | 2952 (88) | 344.46 |
| 2009 | 26.99 | 263.76 | 265.04 | 5257 (97) | 349.00 |
| 2010 | 25.86 | 255.94 | 259.22 | 3714 (91) | 338.88 |
| 2011 | 25.51 | 251.6 | 257.99 | 3657 (82) | 335.82 |
| 2012 | 24.97 | 247.2 | 257.22 | 1542 (71) | 340.57 |
| 2014 | 25.48 | 251.05 | 255.66 | 613 (88) | 329.90 |
| 2015 | 24.58 | 244.21 | 250.41 | 1635 (78) | 314.06 |
| 2016 | 26.01 | 250.18 | 256.81 | 3185 (95) | 341.32 |
| 2017 | 24.56 | 249.91 | 261.71 | 2167 (74) | NA |
| 2018 | 24.47 | 249.86 | 265.16 | 2404 (70) | NA |
| 2019 | 25.97 | 254.91 | 262.97 | 2291 (80) | NA |

**Table 2. Selected GAM-based models for the trends in yellow anaconda (*Eunectes notaeus*) harvest parameters measured by the *Programa Curiyú* between 2002–2019 in northeastern Argentina.**

| Parameter | | Intercept | | Smooth (year) | Deviance explained |
|---|---|---|---|---|---|
| Total captures | Est. | 8,16 | Edf | 5,53 | 69,70% |
| | SE | 0,09 | d.f. | 6,67 | |
| | p | < 0.001 | p | < 0.05 | |
| Proportion of captures in July | Est. | -0,02 | Edf | 5,9 | 63,40% |
| | SE | 0,01 | d.f. | 5,99 | |
| | p | < 0.05 | p | < 0.001 | |
| Proportion of females | Est. | 1,41 | Edf | 5,72 | 64,10% |
| | SE | 0,01 | d.f. | 5,96 | |
| | p | < 0.001 | p | < 0.001 | |
| Number of hunters | Est. | 5,02 | Edf | 2,24 | 85,70% |
| | SE | 0,08 | d.f. | 2,79 | |
| | p | < 0.001 | p | < 0.001 | |
| Hunting days | Est. | 4,62 | Edf | 2,12 | 58,40% |
| | SE | 0,07 | d.f. | 2,64 | |
| | p | < 0.001 | p | < 0.01 | |
| Skin SVLs | Est. | 5,54 | Edf | 2,95 | 79% |
| | SE | 0,002 | d.f. | 2,99 | |
| | p | < 0.001 | p | < 0.001 | |
| Skin SVLs > 230 cm | Est. | 5,56 | Edf | 2,95 | 71,40% |
| | SE | 0,002 | d.f. | 2,99 | |
| | p | < 0.001 | p | < 0.001 | |
| Proportion of SVLs > 230 cm | Est. | 1,94 | Edf | 5,92 | 52,90% |
| | SE | 0,01 | d.f. | 5,99 | |
| | p | < 0.001 | p | < 0.001 | |
| Total CPUE | Est. | 0,27 | Edf | 2,41 | 70,80% |
| | SE | 0,02 | d.f. | 2,91 | |
| | p | < 0.001 | p | < 0.001 | |
| July CPUE | Est. | 0,39 | Edf | 3,6 | 86,90% |
| | SE | 0,01 | d.f. | 3,91 | |
| | p | < 0.001 | p | < 0.001 | |

## Effects of hunting effort and environmental traits on the parameters of the harvested population

Total captures were influenced by hunting effort, as well as mean temperature (Table 3). Mean temperature during the hunting season varied between 17.4 and 21.2 ºC. Total number of captures decreased in the years when mean winter temperature was higher than 19 ºC (Fig 5a). Total captures reached its maximum during the years when hunting effort was close to its mean, and decreased in years of both high and low hunting effort (Fig 5b). Mean temperature during the most productive month (July) was significantly lower than in the rest of the hunting season (t = -3.37, df = 23.94, $p < 0.01$), and varied between 14.9–21.5 ºC (Fig 6). Additionally, the increase in CPUE in the most productive month compared to annual CPUE (t = 4.36, df = 26.79, P < 0.001) was significantly influenced by lower temperatures (Fig 6; GLM -0.03, range: -0.06 to -0.01, $p = 0.004$, deviance (%) = 25.5).

The mean SVL of skins was positively influenced by the hunting effort, but it was not affected by mean temperature (Fig 5c; Table 3). On the other hand, the mean SVL of giant

**Table 3. Selected GLM-based interaction models for yellow anaconda (*Eunectes notaeus*) harvest parameters measured by the *Programa Curiyú* between 2004–2019 in northeastern Argentina.**

| Explanatory variables | Predictor variables | coef. (range) | *p*-value | Deviance explained | AIC |
|---|---|---|---|---|---|
| Total captures | mean temperature | -0.08 (-0.18, 0.02) | < 1 | 86.6% | 240.62 |
| | mean temperature$^2$ | -0.08 (-0.18, 0.005) | < 0.1 | | |
| | hunting effort | -0.75 (-1.15, -0.34) | < 0.001 | | |
| | hunting effort$^2$ | -3.86 (-4.9, -2.8) | < 0.001 | | |
| Mean SVL | hunting effort | 0.04 (0.02, 0.07) | < 0.01 | 42.1% | -75.77 |
| Skin SVL > 230 cm | mean temperature | -0.01 (-0.01, 0.0001) | < 0.1 | 60.7% | -84.56 |
| | mean temperature$^2$ | -0.008 (-0.02, 0.001) | < 1 | | |
| | Year | -0.06 (-0.02, 0.003) | < 1 | | |
| | year$^2$ | 0.02 (0.01, 0.03) | < 0.01 | | |
| SVL giant skins* | mean temperature | -0.02 (-0.03, -0.01) | < 0.001 | 78.5% | -63.44 |
| | mean temperature$^2$ | -0.02 (-0.03, 0.01) | < 0.01 | | |

Model selection was based on the lowest AIC value.

*SVL of giant skins was calculated for the period between 2004 and 2016.

skins, as well as mean skin SVL for snakes > 230 cm were affected by mean winter temperature, decreasing during the years when mean temperature was higher (Fig 5d–5f; Table 3). The proportion of captured females was not influenced by mean temperature (S9 Fig in S1 File). Finally, while the maximum measured water level varied from 0.5 to 1.9 m, it did not have any effect on recorded yellow anaconda harvesting parameters (*p* > 0.05).

## Discussion

While wildlife management has been perceived as a key tool for sustainable use of Neotropical resources [74, 75], few studies have been published on the subject to date [16, 76]. In this study, we have shown that: 1) the number of hunters, hunting season duration, and total number of harvested snakes decreased since 2002; 2) capture per unit of effort has increased; 3) annual captures and size of captured anacondas are influenced by winter temperatures; and, most importantly, 4) harvesting is not negatively regulating the parameters of the anaconda population. Thus, the management plan has proven its usefulness and applicability in a low-cost context, with a hard-to-sample species, leading to its sustainable exploitation.

Snakes have secretive habits and estimating their population parameters requires expensive and time-consuming field work. That way, data obtained from management programs represents a viable source of inference, allowing us to identify safe levels of exploitation [39, 49, 59]. Over time, a decrease in the total number of harvested individuals, changes in size and sex ratio, or an increase in hunting effort are appropriate indicators of overexploitation [39, 59, 60]. The continuous monitoring of these factors must be performed to ensure the harvest remains sustainable.

A study analyzing the management of reticulated pythons (*Malayopython reticulatus*) in Indonesia argues that the minimum sample required for statistical detection of population changes would be 551 harvested individuals [59], an unlikely feat for most field studies. Additionally, the authors suggest that an annual decline of 5%, using samples sizes of 5,000 individuals, would reveal population overexploitation. Declines greater than 10% using smaller samples sizes—of about 1,000 individuals—would also indicate overexploitation [59]. In our study—by means of a sample of 64,176 individuals across nearly 20 years—we conclude that

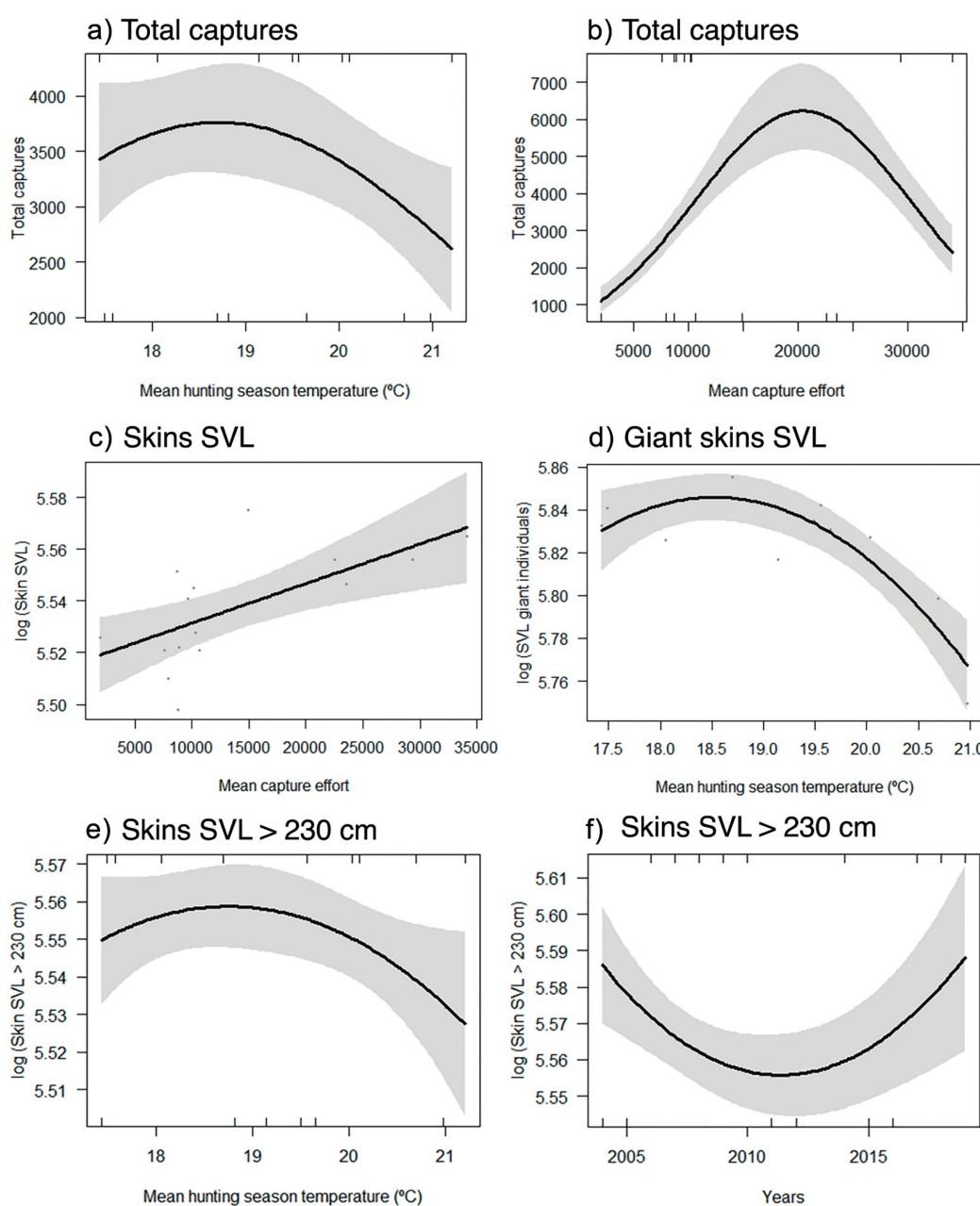

**Fig 5. Effects of the explanatory variables on the parameters of the population of yellow anacondas (*Eunectes notaeus*) harvested by the *Programa Curiyú* between 2004–2019 in northeastern Argentina.** Models selected based on the lowest AIC values are shown. The grey area indicates the 95% confidence interval.

yellow anaconda harvest is sustainable, since the harvesting is not negatively regulating the analyzed parameters (Fig 4c and 4d).

Previous studies suggest that yellow anacondas are resistant to intense exploitation [34, 35], as are pythons in Indonesia [59], tegu lizards (*Salvator* spp.) in Argentina [77] and alligators in the USA [78]. These authors point out that the reasons for this resistance are: 1) high reproductive rates, 2) wide distributions, 3) strong population inter-connectivity, 4) existence of no-hunting areas, 5) exploitation of individuals in disturbed habitats, and 6) low impact hunting

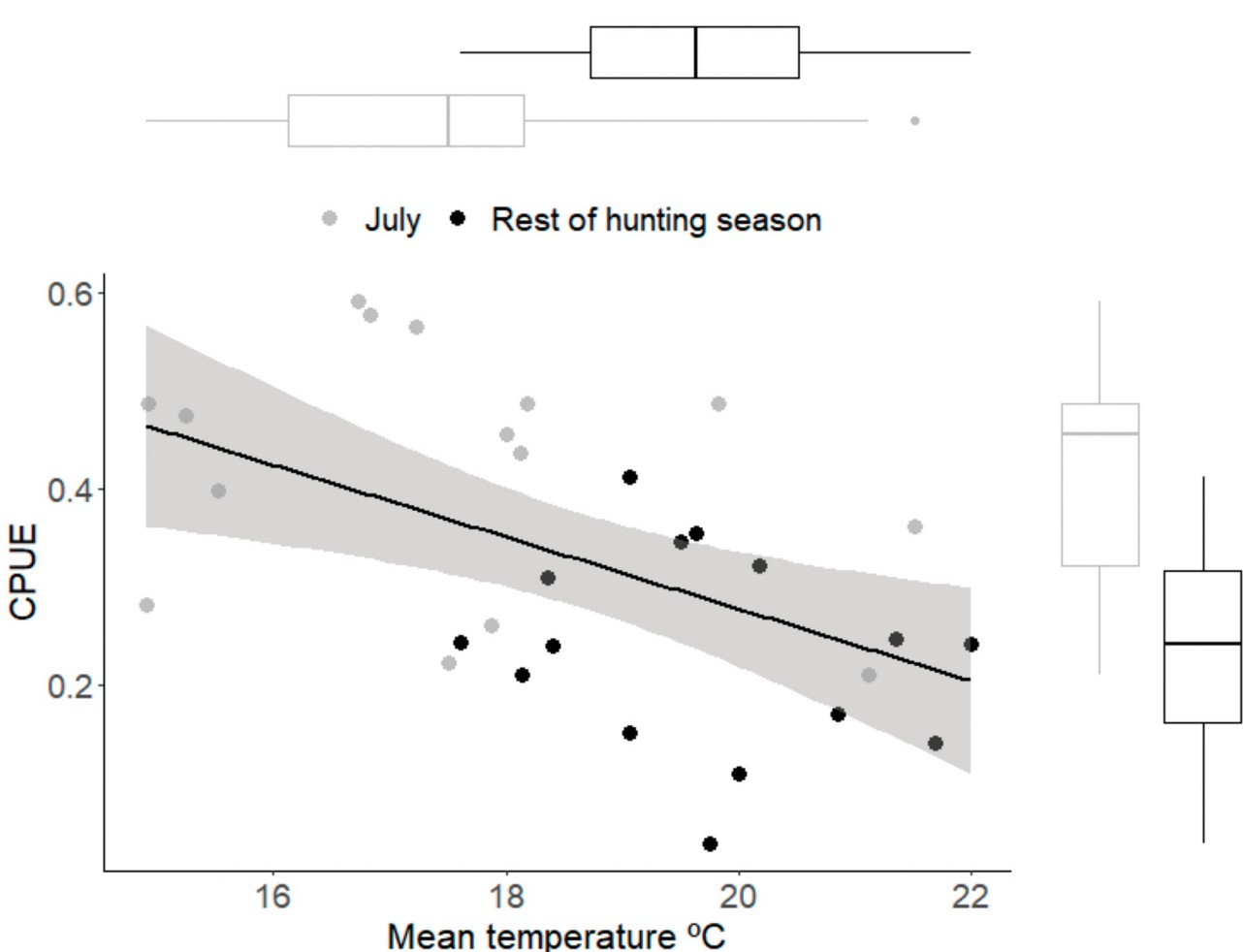

**Fig 6. Capture rate (CPUE) in relation to mean temperature (ºC) for the most productive month (July) and for the rest of the hunting season.** The Gaussian regression for captured taxa (CPUE = 1.01–0.03*mean T ºC) is shown.

techniques (visual search and manual capture). All these conditions are met in the studied population. Yellow anacondas mature at 2.5 years of age with a total length of 2 m, and produce on average of 24 offspring (up to 42; Waller et al. 2007), allowing rapid recruitment of new individuals in a short period of time. In addition, juvenile mortality in anacondas is relatively low due to their large size at birth (40–59 cm, 61–135 g), and to their aquatic, very cryptic and aggressive habits [35]. The species range occupies around 42 million hectares encompassing four countries [35, 57], which makes it unlikely that local harvesting may represent a major threat or cause the extinction of the species. Even so, the anaconda hunting program must continue to be evaluated to ensure it sustainability is kept in the long term.

La Estrella marsh covers around 3,000 km$^2$, and this area represents only 0.7% of the species distribution. The access to the wetland is restricted for hunters due to factors such as water depth, dense floating vegetation, muddy soils, and fallen trunks (Brown et al. 2010; Fig 7). Consequently, large portions of the habitat used by anacondas at La Estrella marsh are not accessible, and this might create a source-sink dynamic [35]. Finally, the technique used to hunt snakes (manual capture) is unlikely to be robust enough to have a significant impact on local anaconda population.

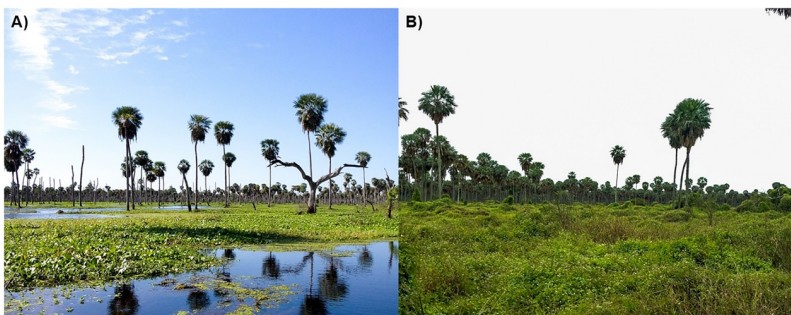

**Fig 7. Different landscape characteristics at La Estrella marsh.** A) Navigable area, suitable for hunting yellow anacondas (*Eunectes notaeus*). B) Area of difficult access where yellow anacondas are not hunted. Photos by B. Camera.

The observed decrease in total annual captures—related to the reduction in hunters and hunting effort—contrasts sharply with the increasing trend in CPUE (Fig 4c). Hunters can be divided in two types: the ones that hunt for extra income, and the ones whose main income comes from hunting. The reduction in hunting effort happened mainly due to the abandonment of the activity by people who were not exclusively dedicated to anaconda hunting [58]. Understanding that there are dedicated and occasional hunters is critical for the correct interpretation of our data, and to prevent inappropriate interventions and population declines of the target species. Skin prices are raised every year to stimulate hunting activities, keeping locals interested on the management program. Finally, the increase in CPUE while hunting effort declined suggests that the anaconda population is not declining in La Estrella.

For snakes in general, long-term analysis of the sex ratio of a population is key for understanding the effects of harvesting programs [39]. Females are a critical factor in managed populations. Harvest programs focused on non-reproductive males would be more appropriate overall [59]. However, snakes of the genus *Eunectes* copulate in aggregations where one female is inseminated by several males. A reduction in the number of males increases the number of infertile eggs in females [15]. Thus, male harvest may have unexpected effects and unintended consequences on the reproductive biology of the species.

Historically, specimens with SVL above 135 cm were harvested [34, 49], indicating that large numbers of non-reproductive individuals were captured. The implementation of a minimum skin size of the 230 cm led to a reduction of 50–60% in production volume [49]. At first, this rule may seem detrimental to the profitability of hunters, but larger skins are wider and have bigger scales, and are therefore pricier then these harvested historically [49].

We emphasize that the size reduction observed in our analysis is an artifact of the data from 2014 and 2015, when the size of giant individuals decreased (see S9 Fig in S1 File) as a consequence of the catastrophic drought of 2013. Measurements returned to previous mean values after 2016. Giant individuals are more likely to die from overheating because they are prone to mobility issues when attempting to leave drying mud pools [15]. This is likely to have caused the widespread mortality of large sized females.

Finally, there is one possible problem for managed harvest we would like to discuss in detail: illegal harvesting. There is no poaching happening at a relevant scale. The program is the only initiative that manages the species, and skins available for international trade need to be labeled as legally derived from the managed population in Argentina. Historically, anaconda populations exploited without the guide of a management program tended to produce small leathers, peaking around 160–180 cm [34], becoming easy to recognize as illegal and

having a lower value for the buyer. Local indigenous populations in La Estrella rightfully hunt a few anacondas to produce trinkets and handicrafts for personal use, and the species is frequently killed when preying over domestic livestock [79], but none of these issues are impacting the demography of the species.

Overall, our data corroborates the hypothesis that the exploitation of yellow anacondas in northeastern Argentina is sustainable. We conclude, therefore, that the parameters chosen by the *Programa Curiyú* for restricting hunting activities are successful in ensuring the demographic viability of the species. The regulation and supervision of management practices by competent institutions and authorities are capable of bringing not only monetary and social benefits to traditional communities, but also to the exploited species by monitoring and conserving its population and habitat. Moreover, wildlife management plans represent a great opportunity for the scientific exploration of many biological aspects of otherwise difficult-to-survey species, such as the yellow anaconda. The harvest must continue to be monitored and assessed to ensure it remains sustainable. Finally, the program exemplifies the use of powerful and cheap tools for sustainable wildlife management, which can be developed for other anaconda populations, as well as for other species and regions.

## Supporting information

**S1 File.**
(DOCX)

## Acknowledgments

We thank the Ministerio de la Producción y Ambiente de la Provincia de Formosa, Fundación Biodiversidad and the Programa Curiyú, Office Vétérinaire Federal of Switzerland, Japan International Cooperation Agency, the Japan Wildlife Research Center, and all hunters for their logistical help.

## Author Contributions

**Conceptualization:** Bruno F. Camera, Itxaso Quintana, Christine Strüssmann, Everton B. P. Miranda.

**Data curation:** Bruno F. Camera, Itxaso Quintana, Tomás Waller, Mariano Barros, Juan Draque, Patrício A. Micucci.

**Formal analysis:** Itxaso Quintana, Everton B. P. Miranda.

**Investigation:** Bruno F. Camera, Christine Strüssmann, Everton B. P. Miranda.

**Methodology:** Itxaso Quintana, Everton B. P. Miranda.

**Project administration:** Bruno F. Camera, Christine Strüssmann, Everton B. P. Miranda.

**Supervision:** Bruno F. Camera, Christine Strüssmann, Everton B. P. Miranda.

**Validation:** Tomás Waller, Mariano Barros, Juan Draque, Patrício A. Micucci.

**Writing – original draft:** Bruno F. Camera, Itxaso Quintana, Christine Strüssmann, Everton B. P. Miranda.

**Writing – review & editing:** Tomás Waller, Mariano Barros, Juan Draque, Patrício A. Micucci.

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
