## [Decision Letter · Decision Letter 0]

1 Jul 2022

PONE-D-22-12738Assessing the sustainability of yellow anaconda (Eunectes notaeus) harvestPLOS ONE

Dear Dr. Camera,

Thank you for submitting your manuscript to PLOS ONE. Your work was assessed by the Academic Editor an two subject expert reviewers. All three of us felt that the manuscript was interesting, topical, and well written and prepared. We commend the authors for their efforts. However,  despite the merit of the work, we feel that it does not fully meet PLOS ONE’s publication criteria as it currently stands. Therefore, we invite you to submit a revised version of the manuscript that addresses the points raised during the review process.

REQUIRED TOPICS TO ADDRESS1. Edit for English language. The manuscript is well written but there are still many instances in which grammar or word choice is not correct. The authors should seek help from a colleague or professional editing service to ensure that the text is error-free.2. Data analysis needs to be clarified. As pointed out by the reviewers, it is sometimes difficult to follow what predictor variables were being analyzed and how model selection was being carried out. AIC-based model selection was described, but the actual selection itself was not shown in the manuscript.3. Justify the inclusion of the pilot study years or remove them. As pointed out by a reviewer, the first two years of data are from a pilot version of the wildlife management program and they may not reflect the general patterns in the rest of the dataset.4. Reduce the strength of conclusions about sustainability. The data analysis in this paper is all based on harvest, which is not necessarily a direct reflection of population size and health, There are many examples from fisheries science in which harvest rates did not seem to be affected, but populations ultimately collapsed. There is a hint of a reduction in the size of females in the data presented, which may suggest that hunting is indeed having some sort of effect. The authors need to be more careful in how they present the "sustainability" of the program.5. Address comments from reviewers. The subject experts provided a lot of excellent feedback. The authors should carefully consider this in their revisions and response.6. Data accessibility. The authors need to ensure that all data necessary to recreate their analyses is freely available.

We look forward to receiving your revised manuscript.

Kind regards,

Christopher M. Somers

Academic Editor

PLOS ONE

Journal Requirements:

2. Thank you for including your ethics statement:  "N/A".   

To comply with PLOS ONE submissions requirements, please provide the following information in the Methods section of the manuscript and in the “Ethics Statement” field of the submission form (via “Edit Submission”):  

*  Please indicate whether an animal research ethics committee prospectively approved this research or granted a formal waiver of ethics approval.*  Please enter the name of your Institutional Animal Care and Use Committee (IACUC) or other relevant ethics board. Also include an approval number if one was obtained.

*   If anesthesia, euthanasia, or any kind of animal sacrifice is part of the study, please include briefly in your statement which substances and/or methods were applied.

For additional information about PLOS ONE submissions requirements for ethics oversight of animal work, please refer to http://journals.plos.org/plosone/s/submission-guidelines#loc-animal-research  

[We thank the Ministerio de la Producción y Ambiente de la Provincia de Formosa, Fundación Biodiversidad and the Progama Curiyú, Office Vétérinaire Federal of Switzerland, Japan International Cooperation Agency, the Japan Wildlife Research Center, and all hunters for their logistical help. We thank the Conselho Nacional de Desenvolvimento Científico e Tecnológico (CNPq) for the financial support through Edital Universal number 003/2014 (process 155536/2014 and 456497/2014–5), and the research fellowship 3123038/2018-1. We thank Fundação de Amparo à Pesquisa do Estado de Mato Grosso (FAPEMAT) for grant conceded in Edital 017/2015.]

 [The author(s) received no specific funding for this work.]

5. We note that Figure 1 includes an image of a participant in the study. 

Reviewers' comments:

Reviewer's Responses to Questions

**Comments to the Author**

1. Is the manuscript technically sound, and do the data support the conclusions?

Reviewer #1: Partly

Reviewer #2: Yes

2. Has the statistical analysis been performed appropriately and rigorously? 

Reviewer #1: Yes

Reviewer #2: Yes

3. Have the authors made all data underlying the findings in their manuscript fully available?

Reviewer #1: Yes

Reviewer #2: Yes

4. Is the manuscript presented in an intelligible fashion and written in standard English?

Reviewer #1: Yes

Reviewer #2: Yes

5. Review Comments to the Author

Reviewer #1: The authors have used nearly 20 years of harvest and demographic data to indirectly assess the viability of commercial anaconda harvest. This is a very important investigation with real conservation implications. I commend the authors for this study and wish we could see more of this in Herpetology. I feel that this is a robust investigation, well-researched, and well-written. Most of my comments are suggestion to improve clarity. However, i do have a couple of larger comments that I hope that the authors will address that will improve the manuscript.

The framework of this manuscript is evaluating the efficacy of an adaptive management program. The authors describe this as an experiment that learns from itself (I like that). However, I ask the authors to please provide more detail about this adaptive management program. What parameters are varied from year to year (presumably the number of permits given out and the length of the hunting season?). However, adaptive management makes its decisions based on observed results - so if the number of permits and the season length vary in response to something - what is it that they vary in response to? And could this changing management influence the results that the authors are looking for? I truly don't know and I think more details about the adaptive management program are required.

The authors include data from two pilot years. These years were extreme outliers in several regards with a hunting season that was about twice as long as all other years and significantly more hunters being given permits. I'm not entirely sure that these years should have been included in the analyses or perhaps analyses should be repeated with and without these years. I suspect that if they are removed many of the key results displayed in Fig 4 and highlighted in the Discussion will be different. How will this alter the conclusions?

Please provide rationale for the demographic parameters of interest earlier in the manuscript. I understand the difficulties in directly studying snake populations (its usually impossible) - thus you relied on these indirect metrics of population health. But because of this - I think more time and attention should be devoted to providing the reader rationale for why these parameters were of interest and how they directly relate to anaconda population viability. The authors describe some of this rationale in the 2nd paragraph of the Discussion but I think it needs to be featured in Introduction to better set up the rationale for the study design. In the Discussion the authors cite reference 47 evaluating harvest of reticulated pythons. The approaches between the two studies are extremely similar but the other study spends considerable time setting up the rationale that because snake abundance is difficult to measure, making assumptions that lack of observed changes to body size and demography of individuals suggests that harvest may be sustainable.

I also think there are places where authors should temper their statements about showing the sustainability of this commercial harvest program and ensure they present their results in the context of these indirect measurements of population viability.

L17: Population viability - not species viability

L22: Demographic parameters not biological

L23-24: I think this should be re-worded to be more cautious. This investigation does not directly evaluate population sustainability but rather uses indirect methods that hint at the maintenance of a healthy population.

L35-36: Annual value of industry? Clarify

L37: delete "nowadays"

L49: delete the comma before "due"

L54-56: I don't understand the point being made here. Clarify. How do wildlife suffer when hunting is outlawed?

L59: citation needed

L67: replace specific with demographic

L76: "Have been". Please state the entity that manages Anaconda. Federal government?

L82: This tolerance might need to be clarified. What does this mean?

L89: I think there should be a table or graph that shows how the adaptive management has changed the parameters of the harvest season in different years.

Lines 90-102 - I think authors would benefit by telling the reader how they relate the 5 measurements they are taking to population sustainability. What are the predictions or how would a change in these values tell us about changing python populations. I believe this should be stated here.

L96: What is rationale for this 230 cm cutoff?

L111: "as of the publication of this report"

L115: indices

L116: delete "managing"

L126-129: re-write for clarity.

L130: delete "an individual of"

L155-160: Suggest providing more biological information about the anaconda that is relevant to population demographics. Age or size at maturity for each sex as well as reproductive capacity (how many clutches per year and how larger are clutches). Also important, when do they reproduce and does this correspond with hunting season?

Equations 1 and 2: where are these derived from and how were they developed?

L247: how could a binomial distribution be used? These are continuous variables.

L270: rephrase. "established" doesn't' make sense

L285: "The average size of captured individuals"

L293-294: Is this a cause for concern?

Table 1: we are seeing a decrease in the percentage of "giant" snakes harvested - this is some cause for concern. Over last 3 years mean % is 75% where in previous years it was high 80 to 90%. Suggests fewer giants around. If larger anacondas have larger clutches, this could have population level effects and implications for recovery. Why the NAs for last 3 years?

Table 2: I do not understand what some of these statistics show.

L360-368 - This section gets very confusing regarding what the sample sizes in each study reference. That citation also highlights the need for decades long measurements.

L368 - although you are seeing a decrease in the giants. I think this is important

L375-381 - really good information that could be more effective if presented in the Introduction (or methods where the Anaconda is first described)

L379-380: good information in paragraph. But is there exploitation in surrounding regions or countries or is this program the only one? Important point.

L386: replace likely with May.

L387: describe methods more

L403: not sure I agree with this statement. Without inclusion of the pilot program - not sure that CPUE would vary across study.

L408-411: I don't know if this means that one male can't successfully inseminate a female

Reviewer #2: This manuscript is well-written and statistical methods seem sound. It would be nice if the authors would include more references to work conducted with the American alligator, which is unarguably the most successful sustained use conservation story in history.

6. PLOS authors have the option to publish the peer review history of their article (what does this mean?). If published, this will include your full peer review and any attached files.

Reviewer #1: No

Reviewer #2: No

---

## [Author Response · Author response to Decision Letter 0]

1 Sep 2022

Specific comments to editors and reviewer are found in the document "Camera et al 2022 Response to Reviewers".

---

## [Decision Letter · Decision Letter 1]

12 Oct 2022

PONE-D-22-12738R1Assessing the sustainability of yellow anaconda (Eunectes notaeus) harvestPLOS ONE

Dear Dr. Miranda,

Thank you for submitting your manuscript to PLOS ONE; the revised version is a substantial improvement. However, there are still several items that require your attention. Therefore, we invite you to submit a revised version of the manuscript that addresses the points raised during the review process.

REQUIRED REVISIONS1. Table 1 - measures of dispersion such as standard deviation such be presented with mean values in this table. Decimal places need to be consistent for all values presented in the table. 2. Sustainability of anaconda harvest - the Academic Editor and subject expert reviewer continue to hold the view that the authors are too firm in their conclusions regarding the sustainability of snake harvest. The authors have not budged at all on this issue in the revisions. To balance the message of the paper, the authors should include a statement in the Abstract and Discussion (around line 354 perhaps?) that indicates that the harvest should continue to be monitored and assessed to ensure it remains sustainable.  3. Small text changes - line 375 what is the size of these anacondas at maturity? Line 428 - "explored" should be changed to "exploited". 

We look forward to receiving your revised manuscript.

Kind regards,

Christopher M. Somers

Academic Editor

PLOS ONE

Journal Requirements:

Reviewers' comments:

Reviewer's Responses to Questions

**Comments to the Author**

1. If the authors have adequately addressed your comments raised in a previous round of review and you feel that this manuscript is now acceptable for publication, you may indicate that here to bypass the “Comments to the Author” section, enter your conflict of interest statement in the “Confidential to Editor” section, and submit your "Accept" recommendation.

Reviewer #1: All comments have been addressed

2. Is the manuscript technically sound, and do the data support the conclusions?

Reviewer #1: Yes

3. Has the statistical analysis been performed appropriately and rigorously? 

Reviewer #1: Yes

4. Have the authors made all data underlying the findings in their manuscript fully available?

Reviewer #1: Yes

5. Is the manuscript presented in an intelligible fashion and written in standard English?

Reviewer #1: Yes

6. Review Comments to the Author

Reviewer #1: (No Response)

7. PLOS authors have the option to publish the peer review history of their article (what does this mean?). If published, this will include your full peer review and any attached files.

Reviewer #1: No

---

## [Author Response · Author response to Decision Letter 1]

29 Oct 2022

1. Table 1 - measures of dispersion such as standard deviation such be presented with mean values in this table. Decimal places need to be consistent for all values presented in the table.

R: the values shown reflect the data on the reports made by Fundación Biodiversidad to which we had access. For some cases it is not possible to produce standard deviation because we didn’t had access to the original data, while in other cases we did.

Decimals have been fixed. 

2. Sustainability of anaconda harvest - the Academic Editor and subject expert reviewer continue to hold the view that the authors are too firm in their conclusions regarding the sustainability of snake harvest. The authors have not budged at all on this issue in the revisions. To balance the message of the paper, the authors should include a statement in the Abstract and Discussion (around line 354 perhaps?) that indicates that the harvest should continue to be monitored and assessed to ensure it remains sustainable. 

R: Thank you for the advice, we agree that the message must be balanced. We added the sentence “The harvest must continue to be monitored and assessed to ensure it remains sustainable.” in the manuscript conclusion, since the line 354 it the end of a paragraph and the start of an unrelated table. In addition, we added the sentence “Even so, the anaconda hunting program must continue to be evaluated to ensure it sustainability is kept in the long term.” as a closing sentence of a paragraph discussing the “whys and hows” the resilience of some reptiles to hunting. Finally, we added the sentence “The continuous monitoring of these factors must be performed to ensure the harvest remains sustainable.” in the lines 389-290.

3. Small text changes - line 375 what is the size of these anacondas at maturity? Line 428 - "explored" should be changed to "exploited".

R: We changed the phrase to “Yellow anacondas mature at 2.5 years of age with a total length of 2 m, and produce on average of 24 offspring (up to 42; Waller et al. 2007), allowing rapid recruitment of new individuals in a short period of time.” and added the text change.

---

## [Editor Report · Decision Letter 2]

1 Nov 2022

Assessing the sustainability of yellow anaconda (Eunectes notaeus) harvest

PONE-D-22-12738R2

Dear Dr. Miranda,

We’re pleased to inform you that your manuscript has been judged scientifically suitable for publication and will be formally accepted for publication once it meets all outstanding technical requirements.

Kind regards,

Christopher M. Somers

Academic Editor

PLOS ONE
---

## [Editor Report · Acceptance letter]

21 Nov 2022

PONE-D-22-12738R2 

Assessing the sustainability of yellow anaconda (*Eunectes notaeus*) harvest 

Dear Dr. Miranda:

I'm pleased to inform you that your manuscript has been deemed suitable for publication in PLOS ONE. Congratulations! Your manuscript is now with our production department. 

Kind regards, 

on behalf of

Dr. Christopher M. Somers 

Academic Editor

PLOS ONE